# Trade-Offs between Competitive Ability and Resistance to Top-Down Control in Marine Microbes

Jinny Wu Yang,[a,b] Feng-Hsun Chang,[b] Yi-Chun Yeh,[c] An-Yi Tsai,[d,e] Kuo-Ping Chiang,[d,e] Fuh-Kwo Shiah,[b,d,f] Gwo-Ching Gong,[d,e] Chih-hao Hsieh[b,f,g,h]

[a]Ecology and Evolutionary Biology, University of Michigan, Ann Arbor, Michigan, USA
[b]Institute of Oceanography, National Taiwan University, Taipei, Taiwan
[c]Department of Biological Sciences, University of Southern California, Los Angeles, Calinfonia, USA
[d]Institute of Marine Environment and Ecology, National Taiwan Ocean University, Keelung, Taiwan
[e]Center of Excellence for the Oceans, National Taiwan Ocean University, Keelung, Taiwan
[f]Research Center for Environmental Changes, Academia Sinica, Taipei, Taiwan
[g]Institute of Ecology and Evolutionary Biology, National Taiwan University, Taipei, Taiwan
[h]National Center for Theoretical Sciences, Taipei, Taiwan

**ABSTRACT** Trade-offs between competitive ability and resistance to top-down control manifest the "kill-the-winner" hypothesis that explains how mortality caused by protists and viruses can promote bacterial diversity. However, the existence of such trade-offs has rarely been investigated in natural marine bacterial communities. To address this question, we conducted on-board dilution experiments to manipulate top-down control pressure (protists only or protists plus viruses [protists+viruses] combined) and then applied 16S rRNA gene high-throughput sequencing techniques to assess the responses of each bacterial taxon. Dilution experiments enabled us to measure the top-down-control-free growth rate as the competitive ability and top-down-control-caused mortality as the reverse of resistance to top-down control. Overall, bacterial taxa with higher top-down-control-free growth rates were accompanied by lower top-down-control-caused resistance. Furthermore, competition-resistance trade-offs were stronger and more consistent when top-down control was caused by protists+viruses combined than by protists only. When protists+viruses were diluted, the bacterial rank abundance distribution became steepened and evenness and richness were decreased. However, when protists were diluted, only richness decreased. Our results indicate the existence of competition-resistance trade-offs in marine microbes and demonstrate the positive impacts of such trade-offs on bacterial diversity. Regardless, the strength of the competition-resistance trade-offs and the impacts on bacterial diversity were contingent on whether top-down control was caused by protists+viruses combined or protists only.

**IMPORTANCE** We addressed the "kill-the-winner" hypothesis from the perspective of its principle (the competition-resistance trade-off) in marine bacterial communities incubated *in situ*. Our results supported the existence of competition-resistance trade-offs and the positive effect on bacterial community diversity. The study linked theoretical expectations and complex natural systems and provided new knowledge regarding how top-down controls and competition trade-offs shaped natural bacterial communities.

**KEYWORDS** marine bacterial communities, kill-the-winner hypothesis, top-down controls, dilution experiments, high-throughput sequencing

Address correspondence to Chih-hao Hsieh, chsieh@ntu.edu.tw.

The authors declare no conflict of interest.

Marine microbial food webs, including bacteria, protists, and viruses, dominate marine biogeochemical cycles (1–5) and represent enormous diversity that is critical in determining marine ecosystem functioning (2, 3, 6). Understanding what maintains

this diversity and allows all these species to coexist is a key focus of microbial ecology, and yet, studies mostly focus on primary productivity and abiotic factors, such as hydrological conditions (7–9). In comparison, how biotic interactions within a community or across trophic levels (e.g., top-down control of bacterial communities caused by protists and viruses) shape marine microbial diversity remains less explored (10–12).

Trade-offs have been identified as important biotic mechanisms for species coexistence (13–16). A trade-off occurs when species specialize in a certain trait at the cost of another trait. In this study, we focused on the trade-off between the growth of species and their resistance to top-down control (hereafter termed the competition-resistance trade-off), which has been studied and tested in terrestrial plants and marine animals (17–19) but poorly assessed in marine microbial communities. In microbial systems, the competition-resistance trade-off is also known as the "kill-the-winner" hypothesis (16), which posits that high growth rates in microbes are accompanied by high mortality rates. For example, a higher density of cell surface protein allows bacteria to have higher nutrient uptake and growth rates, but these proteins are also phage receptors that increase bacterial infection rates (20). Motile and chemotactic bacteria have higher motility, which implies high encounter rates with food but also with predators (21). In addition, developing competitive strategies for resources may come at the cost of reduced defense against natural enemies, such as chemical defenses that reduce protist ingestion and assimilation (21), mechanisms that prevent the entry of phage DNA, and abilities to detect and remove viral nucleic acid (22).

The competition-resistance trade-off (i.e., the kill-the-winner hypothesis) describes how inferior competitors can escape from competition exclusion, thereby promoting species coexistence. This trade-off helps explain how top-down control promotes bacterial diversity across various ecosystems (23–28). However, direct evidence quantifying bacterial growth rates and resistance to top-down control is still lacking. Whether top-down control promotes bacterial diversity in natural communities due to this trade-off remains unanswered.

In this study, we coupled dilution experiments and high-throughput-sequencing techniques to estimate taxon-specific top-down-control-free growth rates and mortality rates in response to protist grazing only or a combination of protist grazing and viral lysis (protists+viruses). Protist grazing and viral lysis are the major causes of mortality affecting bacterial diversity and community composition (23, 26, 29–31). With these estimates, we aimed to (i) determine whether bacterial growth rates were negatively associated with their resistance to top-down control (i.e., the inverse of mortality rates), supporting the existence of competition-resistance trade-off, (ii) determine whether top-down control promotes bacterial diversity, as top-down control alleviates competitive exclusion by suppressing highly competitive species, and (iii) identify potential top-down-control-resistant and top-down-control-susceptible bacterial taxa by investigating the response of each bacterial taxon's net growth rate to increased mortality pressures (protists versus protists+viruses).

## RESULTS

**Competition-resistance trade-off in marine bacterial community.** First, we analyzed the competition-resistance trade-off for each experiment. Four of six experiments showed significant negative relationships between top-down-control-free growth rates and resistance to top-down control by protists (Fig. 1A to F). All six experiments had significant negative relationships between top-down-control-free growth rates and resistance to top-down control by protists and viruses combined (Pearson's correlation coefficient, $r \leq -0.47$ and $P \leq 0.001$; linear regression, all $P < 0.05$) (Fig. 1G to L).

Second, we considered the competition-resistance trade-offs of all experiments using linear mixed-effects model (LMM) analysis. We found significant negative relationships between top-down-control-free growth rates and resistance to top-down control by protists and by protists+viruses combined (LMM, both $P < 0.001$) (Table 1).

**Top-down-control effects on bacterial diversity and composition.** We first focused on the relationships between bacterial diversity (species rank abundance distribution

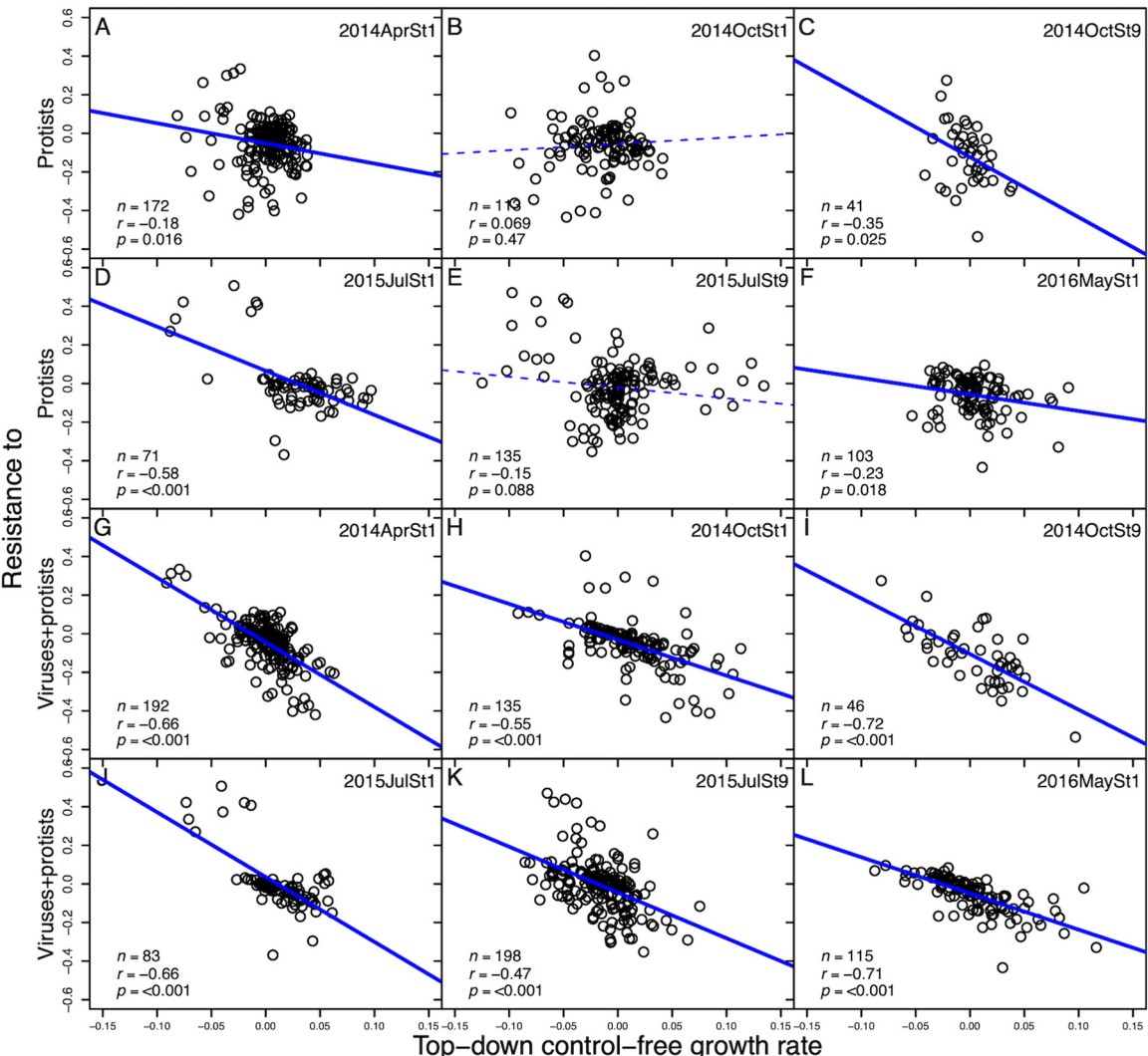

**FIG 1** The relationship between top-down-control-free growth rate ($h^{-1}$) and resistance to protists or protists+viruses-combined effect ($h^{-1}$), respectively, in each experiment. Each point indicates the value for a bacterial ASV. Values in the bottom-left corners represent the sample size ($n$), Pearson's correlation coefficient ($r$), and $P$ value ($p$). Solid and dashed lines indicate significant ($P < 0.05$) and nonsignificant ($P > 0.05$) linear regressions, respectively.

[RAD] decay coefficient, evenness, or richness) and top-down-control dilution factors (TDCF) of all experiments based on LMM analysis. Overall, when top-down control was caused by protists+viruses combined, the RAD decay coefficient decreased and evenness and richness increased with increasing top-down-control dilution factors (LMM; $P < 0.001$) (Fig. 2). However, when top-down control was caused by protists alone, higher top-down-control dilution only increased richness but did not significantly influence evenness and RAD decay coefficient (LMM, richness, $P = 0.013$, and RAD decay coefficient, $P = 0.36$; evenness, $P = 0.33$).

Next, we investigated whether top-down control played a role in maintaining bacterial composition. In general, the compositional distance between the initial samples ($T_0$) and 12-h-incubation samples ($T_{12}$) decreased with increasing top-down-control dilution factors; however, this relationship was only significant in two experiments (October 2014 at station 1 [2014OctSt1] and 2015JulSt1, with linear regression $P$ values of <0.05) under the protists+viruses-diluted treatment (Fig. S3 in the supplemental material). And yet, such a relationship showed no significant difference under protists-diluted and protists+viruses-diluted treatments in all experiments (analysis of covariance [ANCOVA], $P$ values all >0.05).

**TABLE 1** Results of linear mixed-effects model analysis for competition-resistance trade-off and top-down control effects on bacterial diversity indices, with experiments as the random effect

| Independent variable[a] | Dependent variable | Estimate | SE | P value[a] |
|---|---|---|---|---|
| Competition-resistance trade-off | | | | |
| Competitive ability | Resistance to protists | −0.045 | 0.009 | **<0.001** |
| | Resistance to protists+viruses | −0.14 | 0.007 | **<0.001** |
| | | | | |
| Top-down-control effect on bacterial diversity | | | | |
| Protists-diluted | RAD decay coefficient | −0.037 | 0.04 | 0.36 |
| Protists+viruses-diluted | | −0.129 | 0.028 | **<0.001** |
| Protists-diluted | Evenness | 0.013 | 0.013 | 0.33 |
| Protists+viruses-diluted | | 0.04 | 0.009 | **<0.001** |
| Protists-diluted | Richness | 17.96 | 6.91 | **0.013** |
| Protists+viruses-diluted | | 22.23 | 5.33 | **<0.001** |

[a]Bold numbers indicate significant ($P < 0.05$) results.

**Identification of ASVs that are top-down-control resistant or top-down-control susceptible under two top-down-control treatments.** Totals of 89 and 86 amplicon sequence variants (ASVs) had significant responses to top-down control (i.e., significant *per capita* net growth rate versus top-down-control dilution factors [PNGR-TCDF] relationships) in protists-diluted (Fig. 3A) and protists+viruses-diluted treatments (Fig. 3B), respectively. We found that many orders showed a consistent resistant/susceptible pattern to both top-down-control causes. Among them, ASVs belonging to *Vibrionales*, the SAR 86 clade, and *Puniceispirillales* were highly susceptible to both top-down-control causes (i.e., a negative PNGR-TCDF relationship). ASVs belonging to *Gaiellales*, "*Candidatus* Actinomarinales," *Pirellulales*, and *Flavobacteriales* were all or mostly resistant to both top-down-control causes (i.e., a positive PNGR-TCDF relationship). *Rhodobacterales* showed roughly half-and-half resistant and susceptible to both top-down control causes. In contrast, we also found bacterial groups that showed a large divergent response to the two top-down-control causes. For example, *Caulobacterales* and *Ectothiorhodospirales* were strongly resistant to protist grazing but not to the combination of protist grazing and viral lysis. The same analysis including ASVs with nonsignificant responses to top-down control was also conducted, and the results are shown in Fig. S4.

## DISCUSSION

The competition-resistance trade-off (i.e., the kill-the-winner hypothesis) has been widely regarded as an important mechanism for determining microbial assemblages but has rarely been verified in natural marine systems. Previous studies were mostly based on the observation of increased richness or compositional shifts with increased top-down-control intensity (23, 25). Here, we tested the existence of the competition-resistance trade-off by explicitly measuring bacterial growth rates and resistance to top-down control, as well as examining the impact of top-down control on bacterial diversity and composition.

**The existence of competition-resistance trade-offs in marine bacterial communities driven by two top-down-control causes: protists and viruses.** Our findings suggested that bacterial taxa with higher top-down-control-free growth rates had higher mortality rates from top-down control, supporting the presence of the competition-resistance trade-off. In addition, the competition-resistance trade-off was more consistent, stronger, and more significant when mortality was caused by protists+viruses combined than by protists alone. This may be due simply to additive mortality from protists and viruses causing a larger competition-resistance trade-off than protists alone. Another possibility is that viruses are more of a major consumer targeting strongly competitive bacteria than are protists. Marine viruses are, in general, highly host specific (32), and the intensity of bacterium-virus interaction is strongly and positively related to the bacterial growth rate, as reported by a previous study (33). In contrast, marine protists are known to exploit a

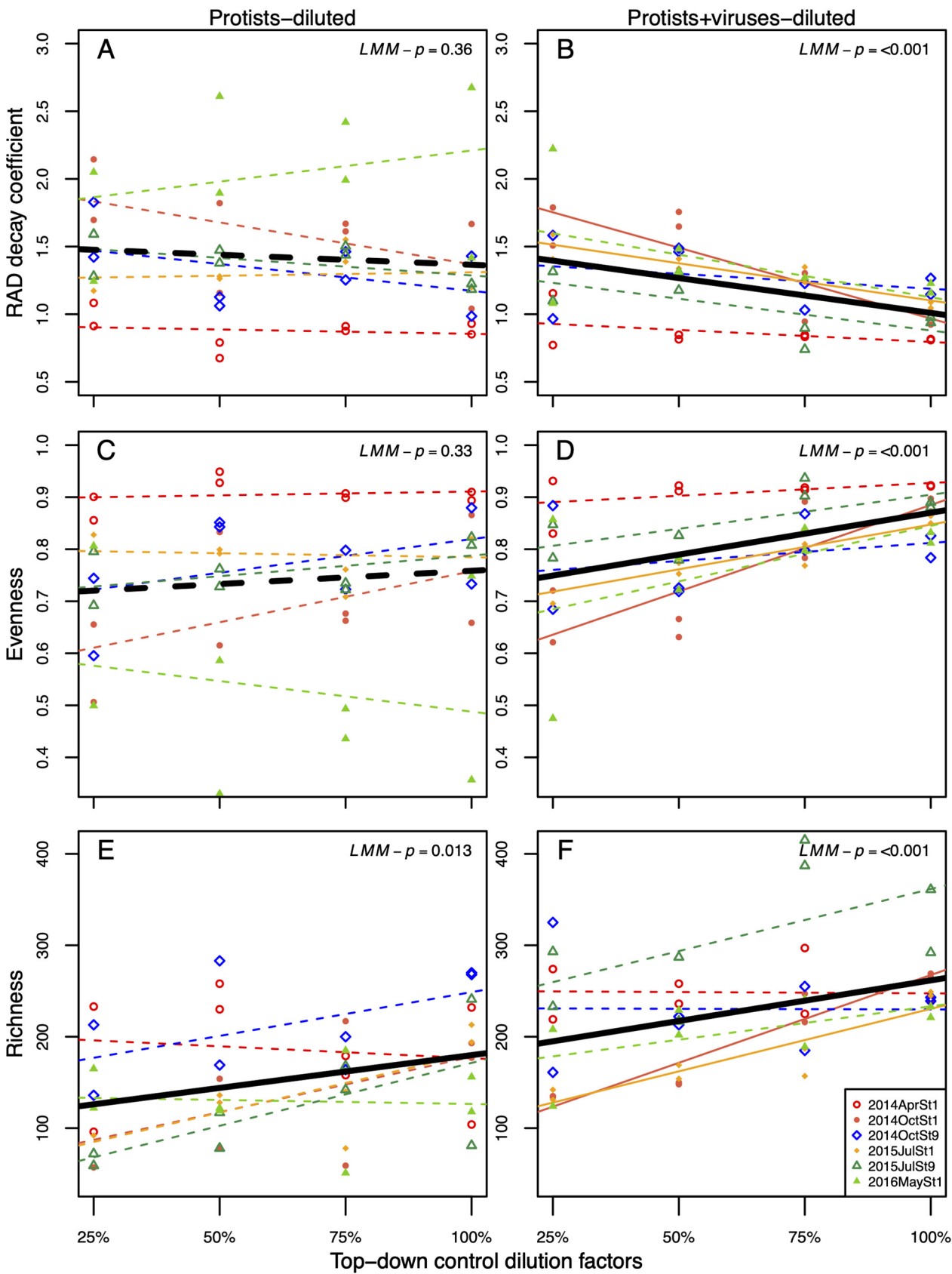

**FIG 2** Relationships between top-down-control dilution factors and bacterial RAD decay coefficient, evenness, or richness at $T_{12}$ under protists-diluted and protists+viruses-diluted treatments. The open red circles, closed orange circles, open blue diamonds, closed dark blue diamonds,

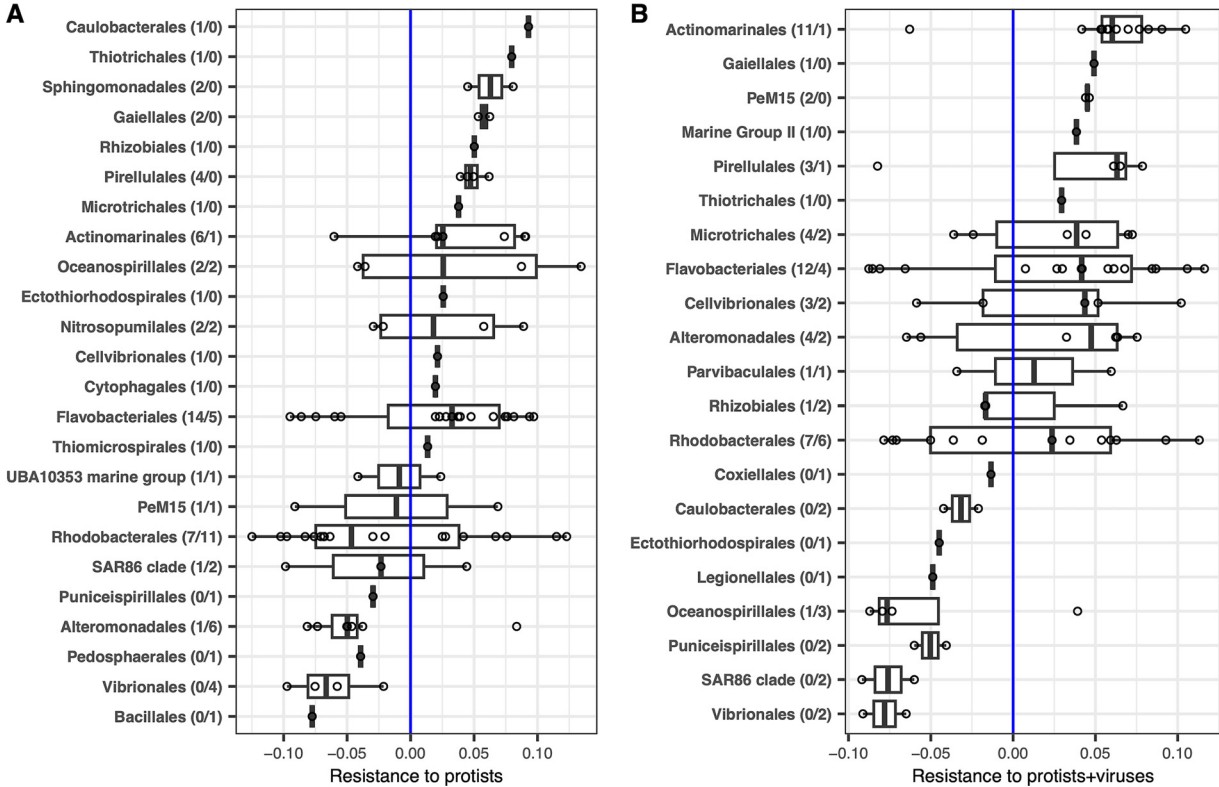

**FIG 3** Bacterial resistance of each ASV classified at the order level. Each point indicates the resistance to top-down control from protists only (A) and protists+viruses (B). A higher positive value indicates a higher resistance to top-down control, whereas a more negative value represents a higher susceptibility to top-down control. Vertical blue lines indicate a resistance value of zero. The numbers in the parentheses after the order name indicate the number of significant resistance ASV/significant susceptible ASV. The order names are ranked from the most resistant (top) to the most susceptible (bottom) to top-down control, based on the mean of ASV resistance to top-down control that the order comprised. The boxes represent the central 50% of data, each with a line inside representing the median. Only ASVs with significant responses to top-down-control dilution factors are shown.

wide spectrum of bacterial species (34–36) and are well-known for their size-based selection (37, 38) which may not strictly focus on the fast-growing "winner."

To test whether viruses were a stronger driver of the competition-resistance trade-off than protists, we subtracted the top-down-control impact of protists+viruses combined from the impact of protists alone in the dilution experiments, following the previous study (39). Based on this method, five of the six competition-resistance trade-offs under virus-caused mortality were stronger than those under protist-caused mortality (Fig. S5). This suggested that viral lysis, most of the time, was the major driver of the kill-the-winner mechanism in marine systems. However, we also note that estimating bacterial resistance to viral lysis using the subtraction method may not be applicable, considering the complex interactions between protists and viruses. For instance, antagonistic interactions between viruses and protists occur when they compete for the same bacterial taxa or when protists feed on free-living viruses or virus-infected bacteria (40–42). In contrast, protist grazing can increase viral infection, although the mechanisms are unclear (43). As a result, bacterial resistance to viral lysis may not be simply estimated by the subtraction method. Experiments manipulating the viral lysis effect only would provide more robust evidence (24, 25) but are technically challenging.

Nevertheless, we found that one experiment suggested protists were a stronger driver of the competition-resistance trade-off than viruses (experiment 2015JulSt1)

**FIG 2** Legend (Continued)

open green triangles, and closed bright green triangles indicate dilution experiments conducted in 2014 April at station 1, 2014 October at station 1, 2014 October at station 9, 2015 July at station 1, 2015 July at station 9, and 2016 May at station 1, respectively. Solid and dashed lines indicate significant ($P < 0.05$) and nonsignificant ($P > 0.05$) linear regressions, respectively. Black bold solid and dashed lines indicate significant (LMM $P < 0.05$) and nonsignificant (LMM $P > 0.05$) linear regressions estimated from the linear-mixed effects model, with experiments as the random effect.

(Fig. S5D). In addition, we found that the initial bacterial community of experiment 2015JulSt1 possessed a much higher relative abundance of *Gammaproteobacteria* than other communities (Fig. S6A). Although the mechanism remains unclear, we anticipate that the initial bacterial community composition plays a role in determining which type of top-down control dominates the kill-the-winner process.

**Linking competition-resistance trade-offs to the impacts of top-down control on bacterial diversity and composition.** Based on the kill-the-winner hypothesis, top-down control promotes bacterial diversity by suppressing superior bacterial competitors and, thus, releasing inferior bacterial competitors from the competitive exclusion (16). While a positive impact of protists or viruses on bacterial diversity has been reported (23, 44), whether this is caused by the competition-resistance trade-off (the kill-the-winner mechanism) remains unverified in natural marine microbial ecosystems. Our findings showed significant competition-resistance trade-offs in bacterial communities and positive top-down-control effects on bacterial diversity when the top-down control was caused by protists+viruses combined. These findings suggested that competition-resistance trade-off is an important driver of positive top-down-control effects on bacterial diversity in marine microbial communities. Although we found a significant competition-resistance trade-off when top-down control was caused by protists alone, the trade-off relationship was weak and less consistent. We also found that dilution of protists did not influence bacterial evenness and RAD. Together, our findings indicated that protists were relatively minor consumers of the fast-growing bacterial competitors compared to viruses and, thus, did not exert a strong enough impact to suppress the "winner" in a bacterial community.

Interestingly, although protists showed little impact on bacterial evenness and RAD, they exerted a significant positive effect on bacterial richness. We anticipated that protists would release bacteria from exclusion via other mechanisms. For example, generalist protists may release dissolved organic matter or be "sloppy eaters" that release nutrients back to the environment when consuming their prey (45). However, further investigation on how protistans are affected by dilution processes and the subsequent influence on bacterial diversity is needed to better understand the impacts of the protist community on bacterial diversity.

Both field and laboratory experiments have provided evidence of an apparent bacterial-composition shift under changing top-down control density, suggesting that top-down control is important in shaping and maintaining bacterial composition (13, 44). If top-down control maintains bacterial composition, we expected that the community under the 100% top-down-control dilution factor (without dilution) would be similar to the original community, which had already been under selection under no top-down-control dilution in the natural marine system. In our case, while we found a better compositional-maintenance effect with increasing top-down-control dilution factors, it was only statistically significant in two experiments (2014OctSt1 and 2015JulSt1) under the protists+viruses-diluted treatment (Fig. S3). Therefore, we inferred that top-down control helped maintain bacterial composition, albeit weakly.

**Linking competition-resistance trade-offs to bacterial taxonomic responses to different top-down-control types.** If protists alone and prostist+viruses combined focused on the same fast-growing bacterial groups, we expected a consistent resistance pattern among bacterial taxa to both types of top-down control. Indeed, we found that most of the bacterial groups that were susceptible/resistant to protist grazing were also susceptible/resistant to the protists+viruses-combined effect. For example, the SAR86 clade and *Puniceispirillales* (SAR116 clade) were susceptible to both top-down-control causes. Both clades are known to predominate in the surface level of coastal marine systems and play a central role in nutrient and carbon cycling (46–48). Active viruses putatively infecting the SAR86 and SAR116 clades have also been observed and studied (49). Furthermore, *Actinomarinales* and *Flavobacteriales* contributed most of the ASVs that were resistant to both top-down-control causes. Both taxa were found to be affiliated with digestion-resistant bacteria that possess specific cell wall properties to prevent them from being digested by protozoa (50, 51). *Actinobacteria* were also often found to be negatively selected by protists (52, 53) and to be slow growing and grazer resistant in marine systems (25). In

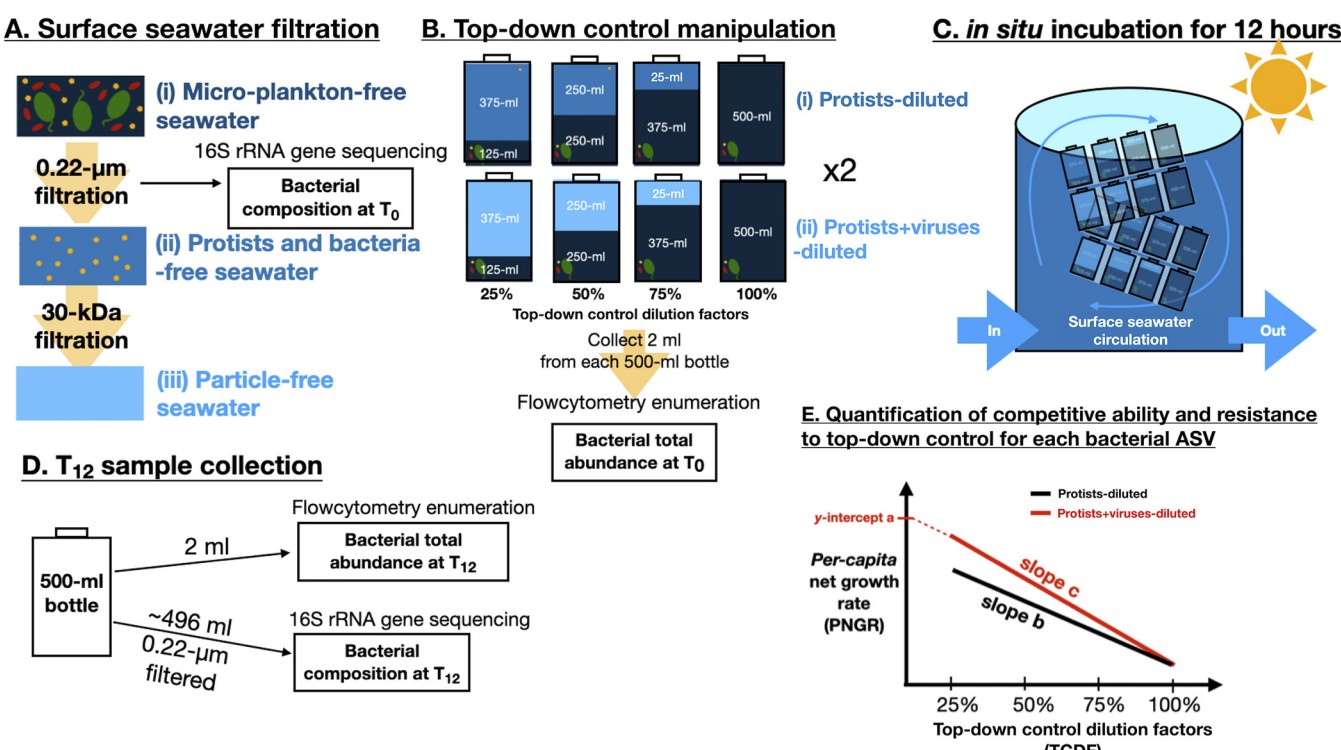

**FIG 4** Illustration of experimental design and sample collection for each top-down-control dilution experiment set. (A) Seawater at the surface layer was collected and sequentially filtered through 20-$\mu$m, 0.22-$\mu$m, and 30-kDa pore-size filters to prepare three types of seawater: (i) microplankton-free, (ii) protist- and bacterium-free, and (iii) particle-free seawater. The 0.22-$\mu$m pore-size filter was collected for estimating bacterial taxonomic composition at $T_0$. (B) These three types of seawater were used for creating (i) protists-diluted and (ii) protists+viruses-diluted treatments in sterilized 500-mL polycarbonate bottles with duplicates. For each treatment, four top-down-control dilution factors were generated, with 25, 50, 75, and 100% of the original top-down-control effect remaining. An aliquot (2 mL) of water was collected from each bottle and used for estimating bacterial abundance at $T_0$. (C) All culture bottles were incubated in a large tank with water flow taken on board from the sea surface layer for *in situ* temperature and without a cover for *in situ* light. (D) After 12 h of incubation, a 2-mL water sample was collected from each bottle for bacterial abundance enumeration at $T_{12}$. The rest of the incubated water from each bottle was filtered on 0.22-$\mu$m pore-size filters for sampling the bacterial community at $T_{12}$. (E) To quantify bacterial competitive ability and resistance, the relationship between *per capita* net growth rate (PNGR) and top-down-control dilution factors (TCDF) for each bacterial taxon was analyzed. Black and red lines represent the PNGR-TCDF regression lines under protists- and protists+viruses-diluted treatment, respectively. Bacterial competitiveness was estimated as the *y* intercept of the PNGR-TCDF relationship under protists+viruses-diluted treatment (*y* intercept a). Bacterial resistance to protist grazing was the PNGR-TCDF regression slope under protists-diluted treatment (slope b). Bacterial resistance to protists+viruses-combined effect was the PNGR-TCDF regression slope under protists+viruses-diluted treatment (slope c).

contrast, we also found some bacterial groups showing different resistance responses to the two types of top-down controls. For example, *Caulobacterales* were resistant to protists but susceptible to the protists+viruses-combined effect, suggesting that these ASVs were resistant to protists but susceptible to viral lysis. Considering that protists and viruses implement different mechanisms (grazing versus infection) to affect bacteria, protists and viruses would target different bacteria with different top-down-control resistance mechanisms (e.g., the ability to escape from grazing versus the ability to detect viral DNA). Overall, our results reflect the complexity of the bacteria-protists-viruses system, not only by being consistent with the existence of the competition-resistance trade-off and how it can be reflected by the bacterial response to top-down controls but also in our finding that bacterial groups showed divergent responses between two types of top-down control.

**Dilution experiments.** Dilution experiments have been used to lower mortality caused by protists and viruses to examine the impact of top-down control on natural aquatic bacterial communities (25, 39, 44, 54, 55). The relationship between the net growth rate of the overall bacterial community and dilution factors is often used to evaluate whether the dilution experiment "works." The expectation is that (i) the net growth rate should decrease with a higher dilution factor and (ii) the regression line of the protists+viruses-diluted treatment should be above the protists-diluted treatment regression line (as shown in Fig. 4E, where the red line is above the black line). This is with the assumption that (i) a higher dilution factor provides higher mortality, which leads to a lower net growth rate of the

overall community, and that (ii) protists+viruses drive higher mortality of the overall bacterial community than do protists alone, where diluting protists+viruses will lead to a higher net growth rate than diluting protists alone. However, these expectations would not be feasible if the competition-resistance trade-off exists in a community and if protists and viruses are targeting different bacterial species, both of which have been observed in this study. This may explain the often-found "unexpected" results in dilution experiments reported by many other studies (54, 55) and ours (Fig. S7). In addition, generally, we found that our dilution process did affect bacterial diversity and cause a directional shift of the bacterial composition along with dilution factors (Fig. S6B), where some bacterial taxa increased while others decreased with increasing dilution factors. This indicates that our dilution of top-down control did have an impact on the bacterial community; however, further study is needed to investigate the reason.

**Conclusion.** Our experiments provided empirical evidence of the existence of a trade-off between competitive ability and resistance to top-down control in marine bacterial communities. Our findings were consistent with the expectations from the kill-the-winner hypothesis: fast-growing bacterial species were less resistant to the combination of protist grazing and viral lysis, as well as to protists alone. In addition, the existence of this competition-resistance trade-off was coupled with a positive top-down-control effect on bacterial community diversity, bridging the gap between the existence of the ecological process (the competition-resistance trade-off) and the consequences for bacterial communities that support the kill-the-winner mechanism. When examining top-down-control-susceptible and -resistant bacterial taxa, while most of the taxa showed a consistent resistance/susceptibility between the two top-down-control causes, some were only resistant to protists and were susceptible to protists+viruses combined. In conclusion, we inferred that competition-resistance trade-offs exist in marine microbes and have a role in maintaining bacterial diversity and influencing bacterial composition and that some bacterial taxa are resistant to both of the top-down-control causes (protist grazing and viral lysis), while some are only resistant to one type of top-down control.

## MATERIALS AND METHODS

**Study sites and top-down-control manipulation experiments.** In total, six sets of top-down-control manipulation experiments were conducted in April and October 2014, July 2015, and May 2016 at two stations in the East China Sea (Fig. S1). Both stations are in the coastal region, with high nutrients and primary production (56). Seawater samples were collected from the surface layer (3 to 5 m) using Go-Flo bottles mounted on a conductivity-temperature-depth (CTD)-equipped rosette (Sea-Bird Electronics, Bellevue, WA, USA).

For each set of experiments (Fig. 4), three types of seawater samples were prepared: (i) microplankton-free seawater (20 $\mu$m filtered), (ii) protist- and bacterium-free seawater (0.22 $\mu$m filtered), and (iii) particle-free seawater (30 kDa filtered). These three types of seawater samples were used for creating two top-down-control treatments: (i) protists-diluted (20-$\mu$m-filtered seawater diluted with 0.22-$\mu$m-filtered seawater) and (ii) protists+viruses-diluted (20-$\mu$m-filtered seawater diluted with 30-kDa-filtered seawater). For each top-down-control treatment, a four-point dilution series (i.e., top-down-control dilution factors) consisting of 25, 50, 75, and 100% microplankton-free seawater was generated. This experimental design assumes that dilution decreases bacterial mortality from protist grazing and viral lysis via decreased encounter rates between bacteria and their predators/parasites (25, 39, 54). Each experimental treatment was set up in duplicates and incubated for 12 h in sterilized 500-mL polycarbonate bottles at *in situ* temperature in an incubation tank with water flow taken on board from the sea surface layer and under *in situ* light.

**Sample collection.** For bacterial enumeration, 2 mL of seawater from each 500-mL bottle was collected at the beginning ($T_0$) and after 12 h ($T_{12}$) of incubation and fixed by a paraformaldehyde solution with a final concentration of 0.2% (57). In total, 32 samples were collected (i.e., 2 top-down-control treatments $\times$ 4 dilution factors $\times$ 2 replicates $\times$ 2 time points) from each set of experiments, with 192 samples collected from the six sets of experiments.

The 0.22-$\mu$m filters that were used to create protist- and bacterium-free seawater ($\sim$10 L) were collected to analyze the bacterial composition at $T_0$. After 12 h of incubation, seawater in each 500-mL incubation bottle was filtered onto 0.22-$\mu$m filters to capture the bacterial composition at $T_{12}$. We acknowledge that a smaller amount of filtered seawater for $T_{12}$ can lead to a lower probability of detecting rare species. From each set of experiments, 17 filter samples were collected, including 1 sample at $T_0$ and 16 samples at $T_{12}$ (2 top-down control treatments $\times$ 4 dilution factors $\times$ 2 replicates). A total of 102 filter samples were collected from the six sets of experiments. All seawater and filter samples were frozen in liquid nitrogen onboard and stored at $-80°C$ until processed.

**Bacterial enumeration, DNA extraction, PCR, sequencing, and bioinformatics.** To enumerate bacteria, 2 mL of paraformaldehyde-fixed seawater sample was stained with SYBR green (Molecular Probes, Inc., USA) in the dark for 15 min and bacterial cells enumerated using a FACSAria flow cytometer

(Becton, Dickinson, USA). The increase in total bacterial abundance from $T_0$ to $T_{12}$ in each experiment indicated that bacteria had grown during the 12-h incubation (Fig. S2).

Genomic DNA was extracted from the 0.22-$\mu$m filters using the DNeasy PowerWater kit (Qiagen, Germany) according to the manufacturer's instructions. The V5-V6 region of 16S rRNA genes was amplified with 2-step PCR (18) (see Text S1 for details). One sample failed in PCR amplification and was not further sequenced (the 25% top-down-control dilution factor under the protists-diluted treatment in experiment 2015JulSt1). The PCR products were purified using Agencourt AMPure XP beads (Beckman Coulter, USA). The purified PCR products were quantified using the Qubit double-stranded DNA (dsDNA) broad-range (BR) assay kit (Life Technologies, USA), pooled in equal amounts, and then sequenced on the Illumina MiSeq PE300.

Raw sequences were processed using DADA2 implemented in QIIME2 (version 2018.8) (58) with standard parameters (59). In brief, forward and reverse reads were truncated where mean quality scores were <25, denoised using the DADA2 algorithm, and then merged. Chimeras were identified and removed. Amplicon sequence variants (ASVs) were then classified with a pretrained naive Bayes classifier (60) against the SILVA version 138.1 database (61). Mitochondria and chloroplast ASVs were removed. Singleton ASVs across all experiments were also removed. The ASV table was then rarefied to the shallowest sample (i.e., 3,188 reads) to avoid sampling bias (Table S1).

**Estimates of top-down-control-free growth rate and resistance to the top-down control of each bacterial ASV.** The relationship between *per capita* net growth rate and top-down-control dilution factors (PNGR-TCDF) was analyzed for each bacterial ASV (hereafter, the PNGR-TCDF relationship) for both of the top-down control manipulation treatments (protists-diluted and protists+viruses-diluted) (Fig. 4E). The *per capita* net growth rate ($R$) was estimated using $R = \ln(N_{12}/N_0)/(12\ \text{h})$, where $N_0$ is the initial population size and $N_{12}$ is the population size after the 12-h incubation. This estimation assumes that the population does not experience density dependency and grows exponentially, so $N_t = N_0 \times e^{rt}$, where $N_t$ is the population size at time $t$ (62, 63). The population size of each bacterial ASV ($N_0$ and $N_{12}$) was estimated by multiplying its relative abundance (estimated from 16S rRNA gene sequencing) by the mean bacterial abundance (estimated from bacteria enumeration) from duplicates. All ASVs with a read of zero at $T_0$ or $T_{12}$ were removed from the analysis, implying a possible limitation in detecting rare taxa.

Linear regression was used to assess the PNGR-TCDF relationship under the two top-down-control manipulation treatments (protists-diluted and protists+viruses-diluted). The slope of the PNGR-TCDF relationship under the protists-diluted treatment was the inverse of the bacterial mortality rate from protist grazing, and thus, it was used to represent bacterial resistance to protist grazing, as it indicated the degree of the bacterial growth rate in response to increased protist grazing. The slope of the PNGR-TCDF relationship under the protists+viruses-diluted treatment was the inverse of bacterial mortality from protists grazing and viral lysis, and thus, it was used to represent bacterial resistance to the protists+viruses-combined effect, as it indicated the degree of the bacterial growth rate in response to the increased effect from protist grazing and viral lysis combined. Finally, the bacterial top-down-control-free growth rate was estimated as the $y$ intercept of the PNGR-TCDF relationship under the protists+viruses-diluted treatment.

**Bacterial community diversity indices.** We investigated three bacterial diversity indices: the decay coefficient of species rank abundance distribution (RAD), evenness, and richness. The decay coefficient of RAD and evenness were estimated based on the rank-normalized RAD that was normalized to the lowest rank (lowest number of species among communities at $T_{12}$ in each experiment) (Table S1) and averaged over 1,000 possible normalized RAD values using the RADanalysis package in the R program. This enables a quantitative comparison of RAD structures and evenness among communities with a controlled richness (64). To describe the RAD structure, the RAD decay coefficient was represented by the estimated decay coefficient obtained from fitting the Zipf model to the normalized RADs using the radfit package in the R program. A community with a higher RAD decay coefficient indicates that the frequencies of species decrease dramatically along with rank (a steeper RAD), whereas a lower RAD decay coefficient represents a smaller difference among species frequencies (a flatter RAD). Evenness was estimated using the vegan package in the R program with rank-normalized RAD. Bacterial richness was ASV richness.

**Data analyses.** To investigate the competition-resistance trade-off, first, we used Pearson's correlation coefficient and linear regression between the bacterial top-down-control-free growth rate and resistance to top-down control (i.e., the inverse of mortality rates caused by protist grazing and the combination of protist grazing and viral lysis, respectively). Second, the overall competition-resistance relationships across all six experiments were determined with a linear mixed-effects model (LMM) considering random intercepts using the nlme package in R, with the bacterial top-down-control-free growth rate as an independent variable, resistance to top-down control as a dependent variable, and experiments as the random effect. Additionally, to detect the potential artifactual slope-intercept relationship, a permutation test was conducted (Table S2). This permutation analysis was performed due to the concern that the estimates of slope (resistance to top-down control) and intercept (top-down-control-free growth rate) were from two closely related (for protists-diluted treatments) or same (for protists+viruses-diluted treatments) regression lines. Specifically, for each top-down control effect (protists-diluted or protists+viruses-diluted), a null model was generated using the linear regression estimates with randomly shuffled dependent variables (the resistance to top-down control) across dilution factors while independent variables (top-down-control-free growth rate) remained fixed, and the model was repeated 1,000 times.

To test whether top-down control promoted bacterial diversity, we analyzed the relationships between top-down-control dilution factors and three bacterial diversity indices (RAD decay coefficient, evenness, and richness) after 12 h of incubation under two top-down control treatments (protists-diluted or protists+viruses-diluted). These relationships were also analyzed with LMM analysis, with top-down-control dilution factors as independent variables, diversity indices as dependent variables, and

experiments as the random effect. Effects from protists and protists+viruses were examined from protists-diluted and protists+viruses-diluted treatments, respectively.

To identify top-down-control-resistant and top-down-control-susceptible bacteria, we investigated the PNGR-TCDF relationship of each bacterial ASV. We expected top-down-control-resistant bacterial ASVs to show a positive PNGR-TCDF relationship (i.e., a higher bacterial net growth rate with less dilution under top-down control). Positive PNGR-TCDF relationships may seem counterintuitive, but this is because (i) top-down-control-resistant bacterial ASVs avoid or elude top-down control and (ii) a higher top-down control pressure can reduce their competition pressure through consuming strong competitors (e.g., top-down-control-susceptible bacterial ASVs) and alternately, promote their growth. In contrast, top-down-control-susceptible bacterial ASVs have negative PNGR-TCDF relationships.

Finally, we investigated how bacterial composition similarity between the initial samples and 12-h-incubation samples (Euclidean distance calculated from centered log-ratio transformed communities) varied under the four degrees of top-down control dilution factors with protists-diluted and protists+viruses-diluted treatments. The average Euclidean distance was calculated with 100 times subsampling at 3,188 reads using the vegan package in program (65). In addition, analysis of covariance (ANCOVA) was conducted to compare the regression slopes under protists-diluted versus protists+viruses-diluted treatments to evaluate whether compositional change varied under different top-down control treatments. Details of the data analysis and R scripts are provided online (https://github.com/jinnyyang/competition-resistnace-trade-off-in-marine-bacterial-community).

**Environmental variables.** Environmental variables were included in analyses since they may produce confounding effects that can weaken or mask the competition-resistance trade-off (7–9). Temperature was recorded by the CTD profiler. Nitrite and nitrate concentrations were measured by the pink dye method, and phosphate concentrations were measured by molybdenum blue using standard methods (56). Environmental and biotic conditions during the experiments are summarized in Table S1.

**Data availability.** Raw sequence data were deposited on the NCBI Read Archive (SRA) under accession number PRJNA749329.

## SUPPLEMENTAL MATERIAL

Supplemental material is available online only.
**TEXT S1**, DOCX file, 0.02 MB.
**FIG S1**, TIF file, 1.8 MB.
**FIG S2**, TIF file, 2.8 MB.
**FIG S3**, TIF file, 3.2 MB.
**FIG S4**, TIF file, 2.8 MB.
**FIG S5**, TIF file, 1.9 MB.
**FIG S6**, TIF file, 2.8 MB.
**FIG S7**, TIF file, 2.6 MB.
**TABLE S1**, DOC file, 0.01 MB.
**TABLE S2**, DOC file, 0.01 MB.

## ACKNOWLEDGMENTS

We thank Ching-Wei Hsu for assistance in conducting experiments and sampling, Hon-Tsen Yu for providing facilities and advice on laboratory work, Sara Jackrel and Vincent Denef for comments, and John Kastelic for English editing of the manuscript.

This work was supported by the National Center for Theoretical Sciences, Foundation for the Advancement of Outstanding Scholarship, and the National Science and Technology Council, Taiwan.

We declare that we have no conflict of interest.

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
