## [Reviewer comments · mSystems]

Trade-offs between competitive ability and resistance to top-down control in marine microbes

Jinny Yang, Feng-Hsun Chang, Yi-Chun Yeh, An-Yi Tsai, Kuo-Ping Chiang, Fuh-Kwo Shiah, Gwo-Ching Gong, and Chih-hao Hsieh

Corresponding Author(s): Chih-hao Hsieh, National Taiwan University

Review Timeline:

Submission Date:	October 21, 2022
Editorial Decision:	January 11, 2023
Revision Received:	February 10, 2023
Accepted:	February 13, 2023

Editor: Jack Gilbert

Reviewer(s): Disclosure of reviewer identity is with reference to reviewer comments included in decision letter(s). The following individuals involved in review of your submission have agreed to reveal their identity: Eva Teira (Reviewer #1)

Transaction Report:

DOI: <https://doi.org/10.1128/msystems.01017-22>

January 11, 2023

Prof. Chih-hao Hsieh
National Taiwan University
Institute of Oceanography
Taipei
Taiwan

Re: mSystems01017-22 (Trade-off between competition ability and resistance to predation in marine microbes)

Dear Prof. Chih-hao Hsieh:

I am extremely sorry for the delay, the editor in charge of this paper had to go on leave suddenly and we are picking up the pieces. Please address these minor comments and it is very likely the paper will be accepted without further review.

Jack Gilbert (EIC)

Thank you for submitting your manuscript to mSystems. We have completed our review and I am pleased to inform you that, in principle, we expect to accept it for publication in mSystems. However, acceptance will not be final until you have adequately addressed the reviewer comments.

Preparing Revision Guidelines

Sincerely,

Jack Gilbert

Editor, mSystems

Journals Department
Reviewer comments:

Reviewer #1 (Comments for the Author):

The authors present a new version of a previously rejected manuscript. Overall, the authors have made a great effort to review and significantly improve the manuscript considering all the reviewer's comments. The results are sound and relevant and will contribute to understanding of the role of biological factors on community assemblage. I only have some additional comments and suggestions. The discussion should be revised to avoid repetitive ideas and to expand the discussion on the impact of top-down processes (predation and virus lysis) on different taxa. Some parts of the discussion should be moved to the results section. In addition, the authors should provide information about the community composition in the initial samples, as this may add to explain the variability among the different experiments.

Specific comments

- Line 45. Please revise "in situ incubation on" for English usage.
- Line 49. I suggest changing "niche" to "competition". The use of "niche" here is very confusing and is inconsistent with the whole manuscript.
- Line 60. Primary production is not an abiotic factor. Please revise.
- Line 71. Please revise the sentence, a verb is missing.
- Line 73. Please revise sentence as bacteria do not intake viruses.
- Line 177. Please revise sentence, I guess the authors mean "assess" instead of "access".
- Line 212. Please revise the sentence for clarity. Predator-resistant bacteria are not released from predation in the 100% treatment, they avoid or elude predation.
- Lines 276-277. Delete "table 1", and indicate "r" and "n" (number of data points) for each analysis.
- Line 285. I suggest including in this section also the results described in the discussion (lines 394-407) and rename it as "Predation effect on bacterial diversity and composition.
- Line 291: revise these data as they do not agree with the p-values in figures 3 and 4.
- Lines 301-302 and elsewhere. The use of the term predation for viral lysis is odd. Viruses do not predate bacteria. It is true that in the protist+virus treatment both predation and viral lysis may be causing mortality, and both are considered top-down processes. Thus, I suggest using mortality or top-down process when referring to both predation and viral infection. For example, in line 302 I suggest changing to "...were only resistant to one of the two mortality causes". Please, revise the whole manuscript to avoid incorrect usage of the term predation.
- Lines 333-334. Please revise this fragment for clarity.
- Line 345. Please delete (58).
- Line 355. Please add references for the positive effect of predation on bacterial diversity.
- Lines 363-365. Revise the sentence for clarity.
- Lines 366-367. This last sentence needs more explanation and references. Otherwise it can be deleted as it does not add much to the discussion.
- Lines 369. Do the authors mean here "richness" instead of "diversity"? I suggest deleting the last part of the sentence "...by driving the competition resistance trade-off".
- Line 380. Again, I suggest using the term mortality instead of predation.
- Lines 381-389. This section should be significantly extended. In the current version there is virtually no discussion about resistant and susceptible bacteria. The authors should discuss and relate their result with ecological strategies of different bacterial taxa. This is a novel aspect of the manuscript and deserves much more attention than this short paragraphs that mostly repeat results.
- Line 391-409. This section include mostly results and should be moved to the results section. The discussion on the effect of predation on bacterial composition can be included in the section "linking the existence of "kill-the winner" and predation effects on bacterial diversity".
- Line 413. Please change "completive" to "competition".
- Lines 424-427. These last paragraph is largely repetitive.
- Figure 2. Please add to each plot the number of data points (n).
- Figures 3 and 4. These two figures could be merged in a single figure.

Reviewer #2 (Comments for the Author):

The authors use the Evans et al viral and protist dilution assay to assess susceptibility to predation by these guilds vs the growth rates of prey. They sequence community composition to see who grows (resistant) or not (susceptible). I think this is an interesting paper addressing an interesting question. However, I have some methodological and data-presentation concerns. Having done a similar project for similar reasons, I found that the dilution assay often just doesn't work as it logically should (Figure 1E). Often things die where they should grow and grow where they should die. Often the black line in F1E is above the red line. Could the authors indicate how often the experiments showed data as expected? I know abundances are shown in SF3, but it is a bit hard to interpret.

Secondly, I am confused where all of the data points in F2 came from. I had expected six data points (from six experiments), and cannot figure out where I went wrong.

The paper could be strengthened by making these things clearer and allow the reader to focus on the interesting question that is quite creatively asked here.

Review of manuscript entitled “Trade-off between competition ability and resistance to predation in marine microbes” by Yang et al.

The authors present a new version of a previously rejected manuscript. Overall, the authors have made a great effort to review and significantly improve the manuscript considering all the reviewer’s comments. The results are sound and relevant and will contribute to understanding of the role of biological factors on community assemblage. I only have some additional comments and suggestions. The discussion should be revised to avoid repetitive ideas and to expand the discussion on the impact of top-down processes (predation and virus lysis) on different taxa. Some parts of the discussion should be moved to the results section. In addition, the authors should provide information about the community composition in the initial samples, as this may add to explain the variability among the different experiments.

Specific comments

Line 45. Please revise “in situ incubation on” for English usage.

Line 49. I suggest changing “niche” to “competition”. The use of “niche” here is very confusing and is inconsistent with the whole manuscript.

Line 60. Primary production is not an abiotic factor. Please revise.

Line 71. Please revise the sentence, a verb is missing.

Line 73. Please revise sentence as bacteria do not intake viruses.

Line 177. Please revise sentence, I guess the authors mean “assess” instead of “access”.

Line 212. Please revise the sentence for clarity. Predator-resistant bacteria are not released from predation in the 100% treatment, they avoid or elude predation.

Lines 276-277. Delete “table 1”, and indicate “r” and “n” (number of data points) for each analysis.

Line 285. I suggest including in this section also the results described in the discussion (lines 394-407) and rename it as “Predation effect on bacterial diversity and composition.

Line 291: revise these data as they do not agree with the p-values in figures 3 and 4.

Lines 301-302 and elsewhere. The use of the term predation for viral lysis is odd. Viruses do not predate bacteria. It is true that in the protist+virus treatment both predation and viral lysis may be causing mortality, and both are considered top-down processes. Thus, I suggest using mortality or top-down process when referring to both predation and viral infection. For example, in line 302 I suggest changing to “...were only resistant to one of the two mortality causes”. Please, revise the whole manuscript to avoid incorrect usage of the term predation.

Lines 333-334. Please revise this fragment for clarity.

Line 345. Please delete (58).

Line 355. Please add references for the positive effect of predation on bacterial diversity.

Lines 363-365. Revise the sentence for clarity.

Lines 366-367. This last sentence needs more explanation and references. Otherwise it can be deleted as it does not add much to the discussion.

Lines 369. Do the authors mean here “richness” instead of “diversity”? I suggest deleting the last part of the sentence “...by driving the competition resistance trade-off”.

Line 380. Again, I suggest using the term mortality instead of predation.

Lines 381-389. This section should be significantly extended. In the current version there is virtually no discussion about resistant and susceptible bacteria. The authors should discuss and relate their result with ecological strategies of different bacterial taxa. This is a novel aspect of the manuscript and deserves much more attention than this short paragraphs that mostly repeat results.

Line 391-409. This section include mostly results and should be moved to the results section. The discussion on the effect of predation on bacterial composition can be included in the section “linking the existence of “kill-the winner” and predation effects on bacterial diversity”.

Line 413. Please change “completive” to “competition”.

Lines 424-427. These last paragraph is largely repetitive.

Figure 2. Please add to each plot the number of data points (n).

Figures 3 and 4. These two figures could be merged in a single figure.

Reviewer comments:

Reviewer #1 (Comments for the Author):

The authors present a new version of a previously rejected manuscript. Overall, the authors have made a great effort to review and significantly improve the manuscript considering all the reviewer's comments. The results are sound and relevant and will contribute to understanding of the role of biological factors on community assemblage. I only have some additional comments and suggestions. The discussion should be revised to avoid repetitive ideas and to expand the discussion on the impact of top-down processes (predation and virus lysis) on different taxa. Some parts of the discussion should be moved to the results section. In addition, the authors should provide information about the community composition in the initial samples, as this may add to explain the variability among the different experiments.

[Response] We appreciated these constructive comments, and revised the manuscript as suggested. We revised the Discussion, extended more discussion on bacterial responses in two types of top-down control among different taxa, and removed the repetitive parts. We also provided the community composition of the initial samples in Figure S6 and discussed this topic in the Discussion in Lines 1104-1110.

Figure S6. Bacterial community composition at (A) T0 and (B) T12 classified at the Order level. The communities were rarefied at 3188 reads. The y-axis indicates the number of reads. In (B), the upper and lower bar plots show the community composition under protists-diluted and

protists+viruses-diluted treatments, respectively. The replicates were taken means. The percentages indicate the top-down control dilution factors.

Specific comments

Line 45. Please revise "in situ incubation on" for English usage.

[Response] This sentence was revised to "*in marine bacterial communities incubated in situ*" in Lines 47-48.

Line 49. I suggest changing "niche" to "competition". The use of "niche" here is very confusing and is inconsistent with the whole manuscript.

[Response] We changed the wording "niche" to "competition" in Line 51.

Line 60. Primary production is not an abiotic factor. Please revise.

[Response] This sentence was revised to "..., yet studies mostly focus on primary productivity and abiotic factors such as hydrological conditions..." in Line 194.

Line 71. Please revise the sentence, a verb is missing.

[Response] This sentence was revised to "*In microbial systems, the competition-resistance trade-off is also known as the "kill-the-winner" hypothesis [16] that describes microbes with high growth rates are accompanied by high mortality rates.*" in Lines 204-207.

Line 73. Please revise sentence as bacteria do not intake viruses.

[Response] This sentence was revised to "...higher density of cell-surface protein allows bacteria have higher nutrient uptake and growth rates, but these proteins are also phage receptors that increase bacterial infection rates" in Lines 207-209.

Line 177. Please revise sentence, I guess the authors mean "assess" instead of "access".

[Response] We changed the word "access" to "assess" in Line 385.

Line 212. Please revise the sentence for clarity. Predator-resistant bacteria are not released from predation in the 100% treatment, they avoid or elude predation.

[Response] The sentence was revised to "*Positive PNGR-TCDF relationships may seem counter-intuitive, but this is because (i) top-down control-resistant bacterial ASVs were avoided or eluded from top-down control and so that (ii) a higher top-down control pressure can reduce competition pressure on them by consuming their competitors (e.g., top-down control-susceptible bacterial ASVs) and alternately, promote their growth.*" in Lines 497-501.

Lines 276-277. Delete "table 1", and indicate "r" and "n" (number of data points) for each analysis.

[Response] Table 1 showed results from LMM analysis while Figure 2 showed results from the linear regression. We, therefore, keep Table 1 in the main text. For better clarification, we revised the result description of "*Competition-resistance trade-off in marine bacterial community*" in Lines 582-593. We also added the number of data points (n) in Figure 2.

Line 285. I suggest including in this section also the results described in the discussion (lines 394-407) and rename it as "Predation effect on bacterial diversity and composition.

[Response] We combined the results of the predation effect on bacterial diversity and composition into the section "*Top-down control effects on bacterial diversity and composition*" in Lines 651-668.

Line 291: revise these data as they do not agree with the p-values in figures 3 and 4.

[Response] we revised the p-values in Lines 659-660.

Lines 301-302 and elsewhere. The use of the term predation for viral lysis is odd. Viruses do not predate bacteria. It is true that in the protist+virus treatment both predation and viral lysis may be causing mortality, and both are considered top-down processes. Thus, I suggest using mortality or top-down process when referring to both predation and viral infection. For example, in line 302 I suggest changing to "...were only resistant to one of the two mortality causes". Please, revise the whole manuscript to avoid incorrect usage of the term predation.

[Response] We changed all the "predation" and "predator" into "top-down control" throughout the manuscript.

Lines 333-334. Please revise this fragment for clarity.

[Response] The sentence was revised to "*Marine viruses are, in general, highly host-specific [44] and the intensity of bacteria-virus interaction is strongly and positively related to the bacterial growth rate as reported by a previous study [45].*" Lines 905-908.

Line 345. Please delete (58).

[Response] We deleted the extra "(58)".

Line 355. Please add references for the positive effect of predation on bacterial diversity.

[Response] The references supporting the positive predation effect on bacterial diversity were added in Lines 1116-1117.

Lines 363-365. Revise the sentence for clarity.

[Response] This sentence was revised and extended for better clarification in Lines 1122-1182.

Lines 366-367. This last sentence needs more explanation and references. Otherwise it can be deleted as it does not add much to the discussion.

[Response] We deleted these sentences as suggested.

Lines 369. Do the authors mean here "richness" instead of "diversity"? I suggest deleting the last part of the sentence "...by driving the competition resistance trade-off".

[Response] This part was removed. The last part of the sentence was deleted as well.

Line 380. Again, I suggest using the term mortality instead of predation.

[Response] We replaced the term "predation" with "mortality" or "top-down control" when describing viral lysis over the manuscript.

Lines 381-389. This section should be significantly extended. In the current version there is virtually no discussion about resistant and susceptible bacteria. The authors should discuss and relate their result with ecological strategies of different bacterial taxa. This is a novel aspect of the manuscript and deserves much more attention than this short paragraphs that mostly repeat results.

[Response] We substantially extend this section and add more discussion about resistant and susceptible bacteria and discuss how it links to our competition-resistance trade-off results in Lines 1203-1372.

Line 391-409. This section include mostly results and should be moved to the results section. The discussion on the effect of predation on bacterial composition can be included in the section "linking the existence of "kill-the winner" and predation effects on bacterial diversity".

[Response] We moved part of this section to the results section titled "*Top-down control effect on bacterial diversity and composition*" in Lines 651-668, and moved the remaining discussion part to the section "*Linking competition-resistance trad-off to the impacts of top-down control on bacterial diversity and composition*" in Lines 1112-1201.

Line 413. Please change "completive" to "competition".

[Response] We changed "completive" to "competition" in Line 451.

Lines 424-427. These last paragraph is largely repetitive.

[Response] We revised this paragraph in Lines 1452-1460.

Figure 2. Please add to each plot the number of data points (n).

[Response] We added the number of data points (the number of ASV in this community considered) for each experiment in Figure 2.

Figures 3 and 4. These two figures could be merged in a single figure.

[Response] We merged Figures 3 and 4 into a single figure as Figure 3.

Reviewer #2 (Comments for the Author):

The authors use the Evans et al viral and protist dilution assay to assess susceptibility to predation by these guilds vs the growth rates of prey. They sequence community composition to see who grows (resistant) or not (susceptible). I think this is an interesting paper addressing an interesting question. However, I have some methodological and data-presentation concerns. Having done a similar project for similar reasons, I found that the dilution assay often just doesn't work as it logically should (Figure 1E). Often things die where they should grow and grow where they should die. Often the black line in F1E is above the red line. Could the authors indicate how often the experiments showed data as expected? I know abundances are shown in SF3, but it is a bit hard to interpret.

[Response] We provided a figure (analogous with Figure 1E) for each experiment as Figure S7. Indeed, the black line is sometimes above or overlapped with the red line. While the net growth rate generally decreased with dilution factors, it is often not statistically significant.

We understand that the expected pattern should be (i) the net growth rate should decrease with a higher dilution factor, and (ii) the regression line of protists+viruses-diluted treatment should be above the protists-diluted treatment regression line. This is based on the assumption that (i) a higher dilution factor provides higher mortality so that leads to a lower net growth rate in the overall community and (ii) protists+viruses drive higher mortality in the overall bacterial community than protists alone, where diluting protists+viruses will lead to a higher net growth rate than protists alone. However, these expectations would not be feasible if the competition-resistant trade-off exists in a community and if protists and viruses are targeting the different bacterial species, which have been observed in this study.

Therefore, we added a paragraph in the Discussion section to discuss this issue and provided an additional Figure S6 to show a compositional shift with increasing top-down control dilution factors, indicating that our dilution experiment did modify the top-down control on the bacterial community. This paragraph is titled "*Dilution experiment*" in **Lines 1374-1445**.

Figure S6. Bacterial community composition at (A) T0 and (B) T12 classified at the Order level. The communities were rarefied at 3188 reads. The y-axis indicates the number of reads. In (B), the upper and lower bar plots show the community composition under protists-diluted and protists+viruses-diluted treatments, respectively. The replicates were taken means. The percentages indicate the top-down control dilution factors.

Figure S7. The relationship between total bacterial community net growth rates versus top-down control dilution factors in each experiment. The black and red lines indicate dilution treatment on protists only and protists+viruses. The solid and dashed lines indicate significant ($p < 0.05$) and non-significant regression lines.

Secondly, I am confused where all of the data points in F2 came from. I had expected six data points (from six experiments), and cannot figure out where I went wrong. The paper could be strengthened by making these things clearer and allow the reader to focus on the interesting question that is quite creatively asked here

[Response] Each point indicates the top-down control-free growth rate and resistance to the top-down control of each bacterial ASV. For better clarification, we revised the Figure 2 legend description in Line 12058 and added the number of points for each experiment (number of ASV analyzed in each experiment). We have substantially revised the presentation of the manuscript, to make things clear to readers. We thank reviewers for the constructive comments that greatly improve the manuscript.

Figure 2. The relationship between top-down control-free growth rate (hour^{-1}) and resistance to protists or to protists+viruses combined effect (hour^{-1}), respectively, in each experiment. Each point indicates the values of a bacterial ASV. Values in the bottom-left corner represent the number of sample size (n), Pearson's correlation coefficient, (r) and its p value (p). Solid and dashed lines indicate significant ($p < 0.05$) and non-significant ($p > 0.05$) linear regressions, respectively.

February 13, 2023

Prof. Chih-hao Hsieh
National Taiwan University
Institute of Oceanography
Taipei
Taiwan

Re: mSystems01017-22R1 (Trade-offs between competitive ability and resistance to top-down control in marine microbes)

Dear Prof. Chih-hao Hsieh:

Your manuscript has been accepted, and I am forwarding it to the ASM Journals Department for publication. For your reference, ASM Journals' address is given below. Before it can be scheduled for publication, your manuscript will be checked by the mSystems production staff to make sure that all elements meet the technical requirements for publication. They will contact you if anything needs to be revised before copyediting and production can begin. Otherwise, you will be notified when your proofs are ready to be viewed.

If you would like to submit a potential Featured Image, please email a file and a short legend to msystems@asmusa.org. Please note that we can only consider images that (i) the authors created or own and (ii) have not been previously published. By submitting, you agree that the image can be used under the same terms as the published article. File requirements: square dimensions (4" x 4"), 300 dpi resolution, RGB colorspace, TIF file format.

We recognize that the video files can become quite large, and so to avoid quality loss ASM suggests sending the video file via <https://www.wetransfer.com/>. When you have a final version of the video and the still ready to share, please send it to mSystems staff at msystems@asmusa.org.

Sincerely,

Jack Gilbert
Editor, mSystems

Journals Department
E-mail: mSystems@asmusa.org